# Do Large Language Models have Lateral Thinking in Puzzle-Solving Games?

## Abstract

Large Language Models (LLMs) show exceptional skills in a wide range of tasks, with their ability in lateral thinking standing out as a particularly intriguing area. Lateral thinking in LLMs allows them to understand deeper or suggested meanings from the context, which is essential for making sense of complex scenarios, especially in puzzle-solving games. To delve deeper into and improve the lateral thinking capabilities of LLMs in the realm of puzzle-solving, we introduce the "Lateral Thinking Puzzles" and construct the accompanying dataset. Our novel $\mathcal{P}$uzzle$\mathcal{V}$erse framework aims to enhance LLMs' lateral thinking in puzzle-solving games. Complementing this, we propose a creativity metric to ensure comprehensive evaluations. Experiments show that the selected LLMs, after being trained with $\mathcal{P}$uzzle$\mathcal{V}$erse, have an average improvement of 101.9% compared to their performance before $\mathcal{P}$uzzle$\mathcal{V}$erse training among all metrics. We also validate the robustness of $\mathcal{P}$uzzle$\mathcal{V}$erse that trained LLMs perform better in other reasoning tasks.

## 1 Introduction

Lateral thinking, first proposed by De Bono (1970), is a creative problem-solving approach that involves looking at situations from unconventional perspectives to make reasoning. It's quite distinct from logic and often more useful in generating creative and effective solutions. Lateral thinking is contrast with vertical thinking, which is the conventional logical process. While the latter is like digging one hole deeper and deeper, the former requires abandoning the hole and striking off to the sidelines to dig numerous experimental holes.

Lateral thinking is important in solving downstream tasks. It encourages us to view problems from various perspectives, leading to more creative solutions. For example, in business management, it helps break traditional thinking patterns, enabling innovative solutions and providing strategic advice that gives companies a competitive edge. In education, cultivating LLMs with lateral thinking abilities allows educators to access tools that foster creative thinking, design engaging learning materials, and encourage students to explore unconventional approaches to problem-solving. In healthcare, lateral thinking can lead to breakthroughs by offering non-traditional diagnostic and treatment suggestions, particularly for rare or complex cases. For instance, Edward Jenner's decision to explore why dairymaids weren't contracting smallpox, instead of why most did, led to the groundbreaking discovery of the smallpox vaccine. Such lateral thinking is also crucial for Large Language Models (LLMs) (Giadikiaroglou et al., 2024). Xie et al. (2023) emphasize lateral thinking is one of the creative thinking process which promote LLMs solve complex problems more effectively. Take the example shown in Fig. 1. When facing the complex scenario where a pilot encounters hydraulic system leakage with no means of replenishing the fluid, the LLM, such as GPT-4 [1], plays the role of vertical thinker that provides traditional suggestions, such as contacting air traffic control, etc. However, the solution

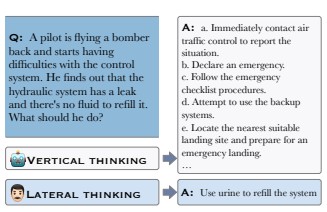

Figure 1: Different solutions given by a vertical thinker (i.e. LLM) and a lateral thinker (i.e. human), respectively, based on a complex scenario.

---

[1] https://chat.openai.com/

given by the human who plays the role as the lateral thinker is using urine which is not a conventional but effective and simple method.

Research on LLMs' lateral thinking in solving downstream tasks is limited. They mainly focus on conventional logical reasoning associated with vertical thinking, which are divided into decomposing tasks and calling external modules. The former includes using Chain of-Thought (CoT) or Auto-CoT to generates reasoning chains (Wei et al., 2022; Zhang et al., 2022), using active learning to stimulate reasoning capabilities (Diao et al., 2023), using a voting strategy to select the most consistent answer output based on different reasoning paths (Wang et al., 2022), etc. The latter includes using frozen LLMs to automatically generate intermediate reasoning steps (Paranjape et al., 2023), decomposing symbolic reasoning, mathematical reasoning, or algorithmic tasks into intermediate steps (Gao et al., 2023), etc. These methods are not enough to make LLMs owe lateral thinking, which necessitates techniques such as challenging assumptions, seeking alternative solutions with analogy, and embracing ambiguity (Xie et al., 2023).

However, lateral thinking varies across different contexts, making the choice of context for studying lateral thinking an important consideration. For instance, the example mentioned above requires external knowledge or commonsense, such as "Urine is mostly water and can substitute for it in emergencies", while some puzzle-solving games demand creativity and imagination, like the riddle "What kind of dog never bites?" and the answer is "A hot dog". Therefore, in this paper, we choose puzzle-solving games to investigate LLMs' lateral thinking which has two main reasons: i) Puzzle-solving games typically require thinkers to step outside conventional thought patterns and apply creativity and imagination to understand and solve puzzles. ii) These games offer a clear framework and objective that is to find the answer to the puzzle. This makes lateral thinking in puzzle-solving games more quantifiable and researchable compared with other more open-ended or subjective scenarios.

To evaluate and enhance LLMs' lateral thinking in puzzle-solving games, we adopt the Lateral Thinking Puzzles (Sloane & MacHale, 1994). Building on the existing lateral thinking puzzles datasets (Jiang et al., 2023; Huang et al., 2023), we construct the largest **L**ateral **T**hinking **P**uzzles dataset (short for "LTP"), which includes riddles, a sequenced set of questions and answers, solutions, and rules. Based on the LTP dataset, we propose $\mathcal{P}uzzle\mathcal{V}erse$ [2], a baseline framework that improves the lateral thinking in puzzle-solving games of LLMs through assisting them to propose a series of questions to clarify the riddle's solution. In addition, we propose a novel creative metric, including compliance, reasoning, and completeness for evaluating LLMs' lateral thinking capabilities. According to the experiments, the $\mathcal{P}uzzle\mathcal{V}erse$ framework can effectively improve LLMs' performance in LTP, resulting in LLMs with advanced lateral thinking in puzzle-solving games. In summary, our study makes three key contributions: i) We construct the largest lateral thinking puzzles dataset. We also propose the creativity metric, adopting it and human metric to evaluate LLMs' lateral thinking in puzzle-solving games. ii) We make an exploration for LLMs' lateral thinking in puzzle-solving games, and then develop a novel $\mathcal{P}uzzle\mathcal{V}erse$ framework to enhance these capabilities in LLMs. iii) We validate the effectiveness of $\mathcal{P}uzzle\mathcal{V}erse$ in LLMs' lateral thinking in puzzle-solving games through extensive experiments in LTP dataset and other reasoning tasks.

## 2 Dataset Construction

In this section, we construct a novel lateral thinking puzzles dataset (abbreviated as "LTP") for evaluating and enhancing LLMs' lateral thinking capabilities in problem-solving games. Each puzzle in LTP comprises a riddle and its corresponding solution. The solutions for riddles in LTP are generally unconventional. As shown in Fig. 2, the riddle states that recently your mother has been acting strangely, often distracted, and sneaking out at night, and you need to discover the truth. The conventional solution is to suspect that the mother is having an affair or involved in some secret activities. However, the unconventional solution is that the mother is participating in square dancing. She sneaks out at night to practice with the team, and to avoid disturbing others, they all dance silently with headphones on. The final solution that she is involved in square dancing does not reveal any secret or suspicious activities.

---

[2]https://anonymous.4open.science/r/haiguitang-EFA7/. We will open-source all data and code after being accepted.

Therefore, due to the unconventional nature of the solutions in LTP, LLMs need to employ lateral thinking without relying on traditional reasoning. They are requested to engage in creative and out-of-the-box thinking to arrive at the solution. Since directly providing a solution based on lateral thinking is highly challenging for LLMs, based on the existing lateral thinking puzzles (Sloane & MacHale, 1994), we set the evaluation of LLMs' lateral thinking capabilities in problem-solving games as follows: for a given riddle, an LLM need to employ lateral thinking through asking yes-or-no questions to infer the solution. An LLM that can infer the solution with the fewer questions is considered to have stronger lateral thinking capabilities in these problem-solving games.

Specifically, we initially collect 647 Chinese lateral thinking puzzles from various websites like Huiwan [3], Baidu Wenku [4], etc. Utilizing GPT-4, we generate additional puzzles that mirror the style and structure but have different semantics from the original ones through in-context learning with the prompt in Table 4 (row "RS Generation"). After generating new puzzles, to ensure that these data points have not been previously learned by the considered LLMs, we remove the original 647 puzzles and use only the generated data for LLMs' evaluation and enhancement. To preserve the unique Chinese characteristics of the dataset and account for the significant semantic differences between Chinese and English, we use the collected Chinese data to expand and create a specialized Chinese dataset. This approach ensures the retention of cultural nuances often lost in translation. Each riddle in the generated puzzles includes only the beginning and end of a story, creating a sense of discontinuity. The solutions require unconventional thinking, differing from standard approaches. Each generated puzzle is assessed using GPT-4 to ensure it meets specific criteria, as detailed in Table 3 (row "RS Evaluation"), with each criterion scored as 0 or 1. Puzzles scoring below 3 are discarded, resulting in a final average score of 3.37.

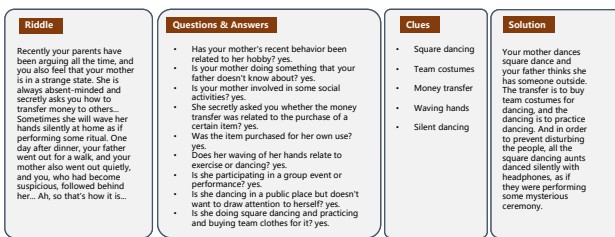

Figure 2: A representative puzzle, which includes a riddle, its solution, questions, answers, and clues.

Subsequently, we employ GPT-4 to create a sequence of questions, answers, and five supporting clues for each puzzle with the prompt in Table 4 (row "QAC Generation"). The questions strictly adhere to a yes-or-no format and are crafted to incrementally lead to the solution, reflecting the unconventional nature of the puzzles. Items with formatting errors are discarded and regenerated. Clues are designed to hint at the solution but not the exact solution, and answers are confined to "yes," "no," or "irrelevant." Each set of questions, answers, and clues per puzzle is also evaluated with GPT-4 to ensure logical progression without significant leaps, adequately hint at the solution, and correctly answer the questions. The criteria are shown in Table 3 (row "QAC Evaluation"), with each criterion scoring 0 or 1. Similarly, sets scoring below 3 are discarded, resulting in a final average score of 3.52. Importantly, for both RS and QAC evaluation, we successively input instructions, such as first asking, "Does the solution require unconventional thinking, differing from standard approaches?" followed by, "Is the overall logic of the puzzle coherent and readable?". This approach migrating the issue where GPT-4, when provided multiple instructions together, may only output partial ratings, such as a single score (e.g., 1) instead of a complete set of scores (e.g., [1,1,1,1,1]).

Table 1: The statistics of LTP.

| Content | Num. |
|---|---|
| Avg. Tokens (Riddles) | 118.4 |
| Max Tokens (Riddles) | 200 |
| Min Tokens (Riddles) | 50 |
| Avg. Tokens (Solutions) | 63.7 |
| Max Tokens (Solutions) | 150 |
| Min Tokens (Solutions) | 30 |
| Avg. Tokens (Questions) | 13.6 |
| Max Tokens (Questions) | 25 |
| Min Tokens (Questions) | 10 |
| Avg. Tokens (Clues) | 4.7 |
| Max Tokens (Clues) | 8 |
| Min Tokens (Clues) | 2 |
| Avg. Number of Rounds | 15.1 |
| Max Number of Rounds | 20 |
| Min Number of Rounds | 7 |

Finally, we make quality validation to ensure the quality and safety of LTP, even with unavoidable themes like suicide and murder. GPT-4 is used to automatically detect and flag potentially unsafe content, discarding entries with detailed descriptions of violence and horror. This process ensures the dataset maintains its integrity while minimizing potential risks associated with sensitive content to the fullest extent possible. Ultimately, we generate a total of 647,000 distinct puzzles. We then

---

[3] https://huiwan.wepie.com/
[4] https://wenku.baidu.com/

Table 2: Comparison of other puzzle-related problem-solving datasets.

| Dataset | Size | Task Type | Language | Difficulty | Evaluation Content | Evaluation Method |
|---|---|---|---|---|---|---|
| BRAINTEASER (Jiang et al., 2023) | 1,119 | Multiple-Choice QA | English | High | Lateral thinking | Model Answering |
| LatEval (Huang et al., 2023) | 325 | Interactive QA | English, Chinese | High | Lateral thinking | Model Asking and Answering |
| Missed Connections (Todd et al., 2024) | 250 | Puzzle Game | English | Medium to High | Puzzle-solving | Model Answering |
| RiddleSense (Lin et al., 2021) | 5,700 | Multiple-Choice QA | English | High | Commonsense reasoning | Model Answering |
| **LTP (Ours)** | **642,600** | **Yes-or-No Questions** | **Chinese** | **High** | **Lateral thinking** | **Model Asking** |

Table 3: Rating criteria for evaluating puzzles in LTP.

| Content | Criteria |
|---|---|
| RS Evaluation | Does the puzzle contain only the beginning and end of a story, creating a sense of discontinuity? If yes, score 1; otherwise, score 0. Does the solution require unconventional thinking, differing from standard approaches? If yes, score 1; otherwise, score 0. Is the overall logic of the puzzle coherent and readable? If yes, score 1; otherwise, score 0. Does the puzzle contain any overly detailed descriptions of violence or horror? If yes, score -100; otherwise, score 1. (-100 means the puzzle is discarded regardless of other scores if detailed negative descriptions are present.) |
| QAC Evaluation | Do the questions strictly adhere to a yes-or-no format? If yes, score 1; otherwise, score 0. Do the questions incrementally lead to the solution with logical coherence and no significant leaps? If yes, score 1; otherwise, score 0. Do the clues hint at but not reveal the solution? If yes, score 1; otherwise, score 0. Are the answers strictly confined to "Yes," "No," or "Irrelevant"? If yes, score 1; otherwise, score 0. |

select 30% of the entries in LTP for manual rating by three volunteers. The criteria for this manual rating combine the first two sets assessed by GPT-4, as shown in Table 3 (rows "RS Evaluation" and "QAC Evaluation"). Puzzles scoring below 6 are discarded, resulting in a final average score of 6.65 and a final count of 642,600 distinct puzzles. To ensure the reliability and validity of the human ratings, we calculate the Inter-rater Agreement using Krippendorff's Alpha and discard data entries with an agreement lower than 0.7, resulting in a final agreement of 0.83. The statistics of LTP are documented in Table 1 and more samples in LTP are shown in Table 1. We also compare LTP with other puzzle-related problem-solving datasets as shown in Table 2, which suggests that the constructed LTP is currently the largest and most comprehensive dataset especially for lateral thinking puzzles.

# 3 Methods

In this section, we introduce $\mathcal{P}uzzle\mathcal{V}erse$, a simple framework inspired by ChatGPT [5] to enhance LLMs' lateral thinking capabilities in puzzle-solving games.

**Supervised Fine-Tuning.** First, we make Supervised Fine-Tuning (SFT) with an LLM. The input consists of riddles, the historical question-answer sequences, and clues with the instruction "Please ask a yes-or-no question based on the riddle [CONTENT], previous question-answer sequences [CONTENT], and clues [CONTENT].", and output the next question. During the training process, we employ scheduled sampling (Bengio et al., 2015) that balances teacher-forcing and free-generation. In the initial stages, teacher-forcing is used to ensure that the LLM

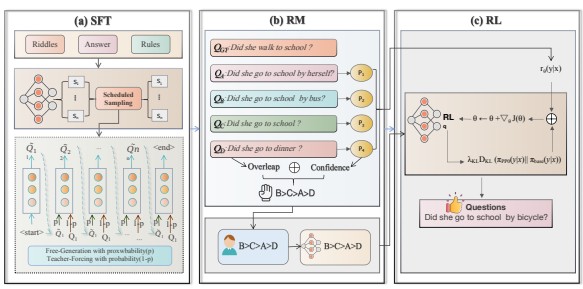

Figure 3: The overview of $\mathcal{P}uzzle\mathcal{V}erse$ framework.

learns the optimal question generation paths. Questions in the training dataset serve as target ones and are used as input to train the LLM in question generation. As training progresses, free-generation is introduced, enabling the LLM to learn to generate questions independently and refine its strategy for progressive questioning. During free-generation, we use the LLM's own generated questions as input and compare these generated questions with the corresponding target question. The proportion of teacher-forcing gradually decreases, and that of free-generation correspondingly increases according to the following equations:

$$p = \frac{1}{1 + e^{-\tau(k-k_0)}}, \quad L_s = pL_t + (1-p)L_f, \quad (1)$$

---

[5]https://chatgpt.com/

Table 4: Prompts for data generation, for the interaction between the questioner LLM and answerer LLM in the inference process, and outputting confidence scores.

| Content | Prompt |
|---|---|
| RS Generation | Given the following puzzle which contains a riddle [CONTENT] and a solution [CONTENT], generate a new puzzle that mirror the style and structure but have different semantics. The generated puzzle contains the riddle and a solution. |
| QAC Generation | The puzzle is [CONTENT]. Given the puzzle, generate a sequence of yes-or-no questions that incrementally lead to the solution. Then generating an answer of each question. The answers is confined to "yes," "no," or "irrelevant." based on the riddle and the solution. After that, provide five supporting clues that hint at the solution without revealing it directly. |
| Questioning | The riddle is [CONTENT]. [The previous questions and answers are [CONTENT]]. Given the riddle, [the previous questions and answers], please ask a "yes-or-no" question. |
| Answering | Please respond to the question in "Yes" or "No" or "Irrelevant". "Irrelevant" means that the current question is not important to deduce the solution. If the answers to five consecutive questions are either "No" or "Irrelevant", provide a clue from the given clues [CONTENT]. You need to give the sign of [SUCCESS] if the questioner deduces the solution within the round limits. Otherwise, you should give the sign of [FAIL]. |
| CS Outputting | Given the following riddle [CONTENT], solution [CONTENT], the question [CONTENT], and the answer [CONTENT], please rate the confidence of the answer on a scale of 1 to 5 (1 being the worst and 5 being the best). |

where $p$ represents the proportion of teacher-forcing, $k$ is the current training step, $k_0$ is the starting step of the decay, $\tau$ is a parameter controlling the decay rate. $L_s$ and $L_t$ represent the respective loss of teacher-forcing and free-generation. $s_1$ to $s_n$ in Fig. 3(a) represent the states.

**Reward Model Construction.** Then, we construct a reward model for the generated questions to encourage LLMs to further generate next questions based on the optimal path. Firstly, we adopt GPT-3.5 as the answerer LLM to answer the generated questions with the prompt in Table 4 (row "Answering"). The questions answered as "Yes" receive positive rewards, while the other questions answered as "No" or "Irrelevant" receive negative rewards. And questions answered as "No" have higher rewards than those answered as "Irrelevant". Subsequently, we determine the overlap score between each positive-rewarded question and the solution. The overlap score measures the similarity, evaluated through sentence embedding using SimCSE (Gao et al., 2021), between the question and the solution. Questions with a higher overlap score receive higher rewards. Additionally, we request the answerer LLM to provide a confidence score between 1 and 5 for the generated questions to further refine the rewards. This confidence score reflects the answerer LLM's trust in its own answers, which is inspired by the reliability metric from LLMs' hallucination evaluation metrics proposed by Chen et al. (2023a) with the prompt in Table 4 (row "CS Outputting").

We then combine the overlap and confidence scores to compute the reward $r_i$ of a generated question $q_i$ as follows:

$$r_i = \begin{cases} \alpha o(q_i) + \beta s(a(q_i)), & \text{if } a(q_i) = \text{Yes} \\ -\alpha o(q_i) + \beta s(a(q_i)), & \text{if } a(q_i) = \text{No} \\ -\gamma \alpha o(q_i) + \beta s(a(q_i)), & \text{if } a(q_i) = \text{Irrelevant} \end{cases} \tag{2}$$

where $o(q_i)$ and $s(a(q_i))$ represents the overlap score and confidence score by the answerer, respectively, for question $q_i$. $\alpha$ and $\beta$ are hyper-parameters in (0,1), and $\gamma$ is a hyper-parameter over 1. This process results in a reliably ranked question sequence $\{q_1, q_2, \ldots, q_{k-1}, q_k\}$ from the most irrelevant to the closest to train a reward model.

Specifically, we adopt an LLM, substituting the softmax layer with a linear layer, to construct the reward model, which receives a generated question sequence as input and outputs a score indicating the question quality. We form pairwise ranking pairs from the ranking sequence's generated questions and utilize the Pairwise Ranking Loss (Liu et al., 2009) for training as depicted below:

$$L_\theta = -\frac{1}{\binom{k}{2}} E_{\sim D}[\log(\sigma(r_\theta(x, y_w) - r_\theta(x, y_l)))], \tag{3}$$

where $x$ represents the original question, $y_w$ and $y_l$ denote the higher-scoring and lower-scoring questions, respectively, in the corresponding ranking pair. $r_\theta$ represents the scalar output of the reward model, $D$ is the set of ranking pairs, and $K$ denotes the number of generated questions. Through this process, the reward model learns to attribute higher scores (rewards) to superior questions and lower scores (rewards) to inferior questions.

**Reinforcement Learning.** After that, we adopt Reinforcement Learning (RL) based on the reward model to further search the optimal question generation path. The state is defined as the riddles, previous question-answer pairs, and clues, with the action being the next question to ask. We employ the Proximal Policy Optimization (PPO) method (Schulman et al., 2017) for training.

# 4 Experiments

In this section, we select some powerful LLMs to explore their lateral thinking capabilities in puzzle-solving games, and further enhance their capabilities with our $\mathcal{P}$uzzle$\mathcal{V}$erse. In this process, an LLM is tasked with formulating questions about a given riddle, then continuing to ask additional questions based on the answers and clues provided by the answerer, who is set to be GPT-3.5.

**Experimental Setups.** We conduct our experiments on four Nvidia A100 GPUs, each with 80GB of memory, using PyTorch in Python. For enhanced training efficiency, we utilize DeepSpeed. We set the maximum sequence length for input and output sequences to 1024 and 200 tokens, respectively. The training process is set to 20 epochs. The detailed configuration of the hyperparameters can be found in Table 5. The prompt of questioning during the inference process is shown in Table 4 (row "Questioning").

During the inference process, we first adopt GPT-3.5 to generate an answer among "Yes", "No", "Irrelevant" for each posed question. The input is comprised of the riddles, questions, and clues, and the corresponding output is the answers to the questions. Secondly, we adopt GPT-3.5 to determine the optimal moment to provide given clues for the questioner LLM. If a question is asked with a positive answer (i.e., "Yes"), it receives positive score (such as plus 1). Conversely, a negative score (such as minus 1) is assigned for the question. If a series of questions consecutively receives negative scores for more than five rounds, GPT-3.5 is then requested to generate a clue to guide the questioner. Finally, the GPT-3.5 determines the questioning's termination. Questioning terminates either when the questioner LLM successfully infers the solution or when the questioning reaches a predefined round limit (we defined it as 30). We further utilize the threshold of the overlap score, which is set as 0.8 tuned through experimentation, to assess the correlation between the sequence of questions and the solution, determining if the solution has been deduced. If the overlap score exceed this threshold within the round limits, it indicates the questioner's successful deduction, prompting GPT-3.5 to declare questioning termination. Alternatively, GPT-3.5 signifies questioning termination when the it reaches the round limits.

Table 5: Parameter configuration and descriptions.

| Parameter Name | Parameter Value | Parameter Description |
|---|---|---|
| Teacher Forcing Ratio ($p$) | 0.8 | The probcapability of using the actual answer as the next input during training, as opposed to using the model's own prediction. |
| Decay Parameter ($\tau$) | 0.9 | Rate at which the teacher forcing ratio decreases over time, allowing the model to rely more on its own predictions during training. |
| Decay Start Step ($k_0$) | 1000 | The training step at which the decay of the teacher forcing ratio begins. |
| Overlap Score Weight ($\alpha$) | 0.7 | Weighting given to the overlap score when determining the relevance of a generated question to its context. |
| Confidence Score Weight ($\beta$) | 0.3 | Weighting given to the confidence score when assessing the quality of a generated question. |
| Penalty for Irrelevant Answer ($\gamma$) | -0.2 | Deductive value applied when a model-generated answer is deemed irrelevant to the context. |
| PPO Clipping Range ($\epsilon$) | 0.2 | Hyper-parameter in PPO that prevents the policy update from changing too drastically, ensuring stable training. |
| Policy Loss Weight ($\mu_2$) | 0.25 | Weight given to the policy loss $L^{clip}(\theta)$ during reinforcement learning training. |
| Value Function Loss Weight ($\mu_3$) | 0.25 | Weight given to the value function loss $L^{VF}(\theta)$ during reinforcement learning training. |

**Datasets, Baselines and Metrics.** LTP is divided into training and validation sets in a 7:3 ratio, with 70% of the data used to train LLMs and the remaining 30% used to evaluate the LLMs' performance. Even without training, the same 30% dataset is used for performance evaluation of the LLMs. We also incorporate other reasoning tasks, similar to lateral thinking puzzles, to validate the effectiveness of LLMs trained with $\mathcal{P}$uzzle$\mathcal{V}$erse. These tasks include story datasets (e.g., LOT (Guan et al., 2022)) and reading comprehension datasets (e.g., DuReader (He et al., 2017), MS MARCO (Nguyen et al., 2016)). The evaluation metrics for these datasets remain consistent with those in the original papers: accuracy for story understanding tasks (i.e., ClozeT, SenPos) and BLEU for story generation tasks (i.e., PlotCom, OutGen) and reading comprehension tasks (i.e., DuReader, MS MARCO).

We choose Baichuan-7B [6], ChatGLM-6B (Du et al., 2022), BELLE-13B (Yunjie Ji, 2023; Yunjie Ji & Li, 2023),MOSS-16B (Sun et al., 2023), and GPT4 as baseline LLMs to evaluate their lateral thinking capabilities. We also adopt $\mathcal{P}$uzzle$\mathcal{V}$erse to enhance the performance of the open-sourced LLMs (the first four LLMs).

To evaluate the quality of the generated questions, we design a comprehensive set of metrics, including creativity metric, machine metric, and human metric. Creativity metric comprises compliance, reasoning, and completeness scores. Machine metric includes BLEU (Papineni et al., 2002), ROUGE (Lin, 2004), the diversity score (Li et al., 2016), and the embedding score (Liu et al., 2016). Human metric is an average score that combines compliance, reasoning, and completeness. Specifically,

---

[6]https://github.com/baichuan-inc/Baichuan-7B

Table 6: Rating criteria by creativity and human for LLMs' generated questions.

| Content | Criteria |
|---|---|
| Creativity Evaluation | Compliance Score: If half or more of the questions in a puzzle are in the yes-or-no format, the score is 1; otherwise, the score is 0.
Reasoning Score: If half or more of the follow-up questions in a puzzle are based on previous information, the score is 1; otherwise, the score is 0.
Completeness Score: If the correct solution to a puzzle is provided within the limited number of turns, the score is 1; otherwise, the score is 0. |
| Human Evaluation | If less than half of the questions in a puzzle are in the yes-or-no format, less than half of the follow-up questions are based on previous question-answer pairs and clues, and the correct solution is not deduced within the limited number of turns, the score is 1.
If half of the questions in a puzzle are in the yes-or-no format, half of the follow-up questions are based on previous question-answer pairs and clues, and the correct solution is not deduced within the limited number of turns, the score is 2.
If more than half of the questions in a puzzle are in the yes-or-no format, more than half of the follow-up questions are based on previous question-answer pairs and clues, and the correct solution is not deduced within the limited number of turns, the score is 3.
If all the questions in a puzzle are in the yes-or-no format, all the follow-up questions are based on previous question-answer pairs and clues, and the correct solution is not deduced within the limited number of turns, the score is 4.
If all the questions in a puzzle are in the yes-or-no format, all the follow-up questions are based on previous question-answer pairs and clues, and the correct solution is deduced within the limited number of turns, the score is 5. |

Table 7: Frameworks related to lateral thinking capabilities.

| Framework | Target Task | Core Technology | Lateral Thinking Support | Innovation | Performance |
|---|---|---|---|---|---|
| Auto-CoT (Zhang et al., 2022) | Logical Reasoning | Automatic Generation of Reasoning Chains | Weak | Traditional reasoning based on logic | Performs well in logical reasoning tasks, but lacks lateral thinking support |
| PAL (Gao et al., 2023) | Algorithmic Reasoning | Automatic Decomposition of Algorithmic Steps | Weak | Focuses on symbolic and algorithmic reasoning | Performs well in mathematical and algorithmic tasks, but not suitable for lateral thinking |
| Connections Solver (Todd et al., 2024) | Puzzle Game | Sentence Embeddings and Instruction-Tuned LLMs | Medium | Combines sentence embeddings with LLMs to solve complex puzzle tasks | Performs well in the "Connections" puzzle task, testing the impact of different prompting styles |
| $\mathcal{P}$uzzle$\mathcal{V}$erse (Ours) | Puzzle-Solving and Lateral Thinking | Question Generation and Reasoning Chain Analysis | Strong | Provides novel evaluation metrics | Excels in the LTP dataset |

the creativity metric is obtained by GPT-4 to assess how well the LLM adheres to the rules and the effectiveness of its generated questions in achieving the solution with 0-1 scale based on the criteria shown in Table 6 (row "Creativity Evaluation"). Scores in this metric are designed based on the characteristics of the lateral thinking game. For instance, the compliance score evaluates whether the generated questions adhere to the basic rules of yes-or-no answers, a critical element in the game. The reasoning score assesses whether follow-up questions are based on previous question-answer pairs. The strength of reasoning ability directly impacts the progress of the puzzle-solving process, making it a crucial evaluation dimension that reflects whether LLMs possess coherent thinking abilities. The completeness score measures the extent to which the generated questions effectively lead to the solution, directly reflecting the effectiveness of LLMs' lateral thinking. Given that the puzzles are designed to be approached from unconventional angles, questions that systematically lead to the solution are considered crucial for fostering lateral thinking. For human metric, we enlist nine human raters to evaluate questions from 1,000 randomly selected puzzles with a 1-5 scale based on the criteria shown in Table 6 (row "Human Evaluation"). The raters kindly offered their assistance without compensation. Inter-rater agreement, measured using Krippendorff's Alpha, is used to ensure rating confidence. Controversial ratings with low agreement (<0.7) are discarded, and questions from another riddle are selected for evaluation. By combining diverse and comprehensive evaluation, we reduce biases that arise from a single evaluation metric, increasing the reliability and credibility of the scoring.

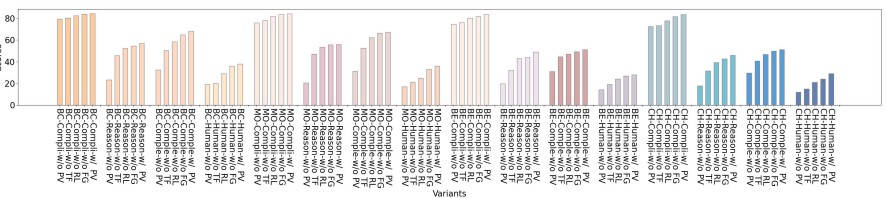

Figure 4: Creativity and human scores of the $\mathcal{P}$uzzle$\mathcal{V}$erse variants, which are removed different modules. BC: baichuan, MO: MOSS, BE: BELLE, CH: ChatGLM.

Table 8: The lateral thinking performance of vanilla LLMs and that of $\mathcal{P}$uzzle$\mathcal{V}$erse-trained LLMs. "PV" means training LLMs with $\mathcal{P}$uzzle$\mathcal{V}$erse.

| | Creativity | | | | | | | | | Machine | | | | | | | | | | | | Human | |
| | Compliance | | | Reasoning | | | Completeness | | | BLEU | | | ROUGE | | | Diversity-2 | | | ES | | | | |
| | w/o PV | w/ AB | w/ PV | w/o PV | w/ AB | w/ PV | w/o PV | w/ AB | w/ PV | w/o PV | w/ AB | w/ PV | w/o PV | w/ AB | w/ PV | w/o PV | w/ AB | w/ PV | w/o PV | w/ AB | w/ PV | w/o PV | w/ PV |
|---|---|---|---|---|---|---|---|---|---|---|---|---|---|---|---|---|---|---|---|---|---|---|---|
| baichuan | 79.5 | 81.3 | **84.4** | 23.4 | 39.5 | **57.0** | 32.3 | 49.1 | **68.1** | 10.9 | 18.6 | **31.1** | 24.3 | 32.6 | 43.5 | 65.8 | 68.2 | **72.9** | 23.5 | 37.6 | **55.0** | 1.9 | **3.8** |
| MOSS | 76.0 | 78.2 | 84.1 | 20.5 | 35.7 | 56.0 | 31.4 | 48.7 | 67.4 | 10.3 | 17.3 | 30.4 | 21.0 | 30.7 | 42.8 | 64.3 | 66.7 | 72.3 | 22.3 | 34.1 | 54.3 | 1.7 | 3.6 |
| BELLE | 74.7 | 77.5 | 83.7 | 19.6 | 28.8 | 48.9 | 31.1 | 46.5 | 51.2 | 9.7 | 16.9 | 30.1 | 29.7 | 33.8 | **49.2** | 62.1 | 65.9 | **72.9** | 21.0 | 30.3 | 53.5 | 1.4 | 2.8 |
| ChatGLM | 72.6 | 76.4 | 83.6 | 17.8 | 25.4 | 46.0 | 29.5 | 43.0 | 51.1 | 10.0 | 16.5 | 30.2 | 19.9 | 28.6 | 40.5 | 61.3 | 65 | 72.8 | 19.5 | 28.7 | 51.1 | 1.2 | 2.9 |
| Average | 75.7 | 78.4 | **84.0** | 20.3 | 32.4 | **52.0** | 31.1 | 46.8 | **59.5** | 10.2 | 17.3 | **30.5** | 23.7 | 31.4 | **44.0** | 63.4 | 66.5 | **72.7** | 21.6 | 32.7 | **53.5** | 1.6 | **3.3** |
| ↑ | - | 2.7 | 8.3 | - | 12.0 | 31.7 | - | 15.8 | 28.4 | - | 7.1 | 20.2 | - | 7.7 | 20.3 | - | 3.1 | 9.3 | - | 11.1 | 31.9 | - | 1.7 |
| ↑(%) | - | 3.5 | 10.9 | - | 59.2 | 155.7 | - | 50.7 | 91.3 | - | 69.4 | 197.8 | - | 32.5 | 85.5 | - | 4.9 | 14.8 | - | 51.4 | 147.9 | - | 111.3 |

Table 9: A comparison of GPT-4 with zero-shot results from other models across 1,000 samples.

| | | Creativity | | | | Machine | | | Human |
|---|---|---|---|---|---|---|---|---|---|
| | Compliance | Reasoning | Completeness | BLEU | ROUGE | Diversity-2 | ES | / |
| baichuan | 77.3 | 22.6 | 35.9 | 11.6 | 27.5 | 64.9 | 24.4 | 1.9 |
| MOSS | 72.4 | 21.5 | 33.1 | 10.1 | 20.5 | 64.1 | 23.5 | 1.8 |
| BELLE | 74.0 | 18.2 | 30.6 | 9.5 | 28.1 | 63.8 | 21.6 | 1.4 |
| ChatGLM | 71.8 | 17.9 | 29.3 | 10.0 | 21.9 | 60.3 | 19.8 | 1.3 |
| GPT4 | **91.7** | **72.5** | **78.8** | **56.2** | **79.3** | **84.4** | **70.1** | **4.3** |

Table 10: One-shot performance of LLMs in other reasoning tasks after being trained with $\mathcal{P}$uzzle$\mathcal{V}$erse.

| | Story Understanding | | | | Story Generation | | | | Reading Comprehension | | | |
|---|---|---|---|---|---|---|---|---|---|---|---|---|
| | ClozeT | | SenPos | | PlotCom | | OutGen | | Dureader | | MSMACRO | |
| | w/o PV | w/ PV | w/o PV | w/ PV | w/o PV | w/ PV | w/o PV | w/ PV | w/o PV | w/ PV | w/o PV | w/ PV |
| baichuan | 81.7 | 88.5 | 70.5 | 78.4 | 29.5 | 34.1 | 51.2 | 59.1 | 49.1 | 58.3 | 42.5 | 47.1 |
| MOSS | 79.3 | 85.4 | 67.5 | 74.6 | 26.3 | 30.7 | 50.4 | 56.2 | 47.5 | 53.9 | 39.7 | 45.9 |
| BELLE | 76.9 | 84.9 | 68.1 | 76.8 | 25.7 | 30.5 | 48 | 55.2 | 47.2 | 54.5 | 38.4 | 45.7 |
| ChatGLM | 76.1 | 83.2 | 64.2 | 70.8 | 23.5 | 27.4 | 45.2 | 53.5 | 46.5 | 52.3 | 38.3 | 44.3 |
| Average | 78.5 | 85.5 | 67.6 | 75.2 | 26.3 | 30.7 | 48.7 | 56.0 | 47.6 | 54.8 | 39.7 | 45.8 |
| ↑ | - | 7.0 | - | 7.6 | - | 4.4 | - | 7.3 | - | 7.2 | - | 6.1 |
| ↑ (%) | - | 8.9 | - | **11.2** | - | **16.9** | - | 15.0 | - | 15.1 | - | **15.2** |

**Main Results.** The lateral thinking performance of vanilla LLMs and that of $\mathcal{P}$uzzle$\mathcal{V}$erse-trained LLMs are shown in Table 8. Results of GPT4 is on 1,000 samples due to resource constraints, and the corresponding zero-shot performance of other baseline LLMs is shown in Table 9. From the initial performance of LLMs (denoted as "w/o PV"), we observe that in compliance, baichuan and MOSS score the highest, while BELLE and ChatGLM score relatively lower. In reasoning, all LLMs score low, with baichuan having the highest score at only 23.4. In completeness, baichuan and MOSS have relatively high scores, whereas other two score lower. Machine metrics show baichuan performing well, while other LLMs also perform similarly overall. In human evaluations, all LLMs have poor performance, with scores not exceeding half. Overall, LLMs' initial lateral thinking capabilities are limited, especially in reasoning and completeness. Moreover, we find GPT-4 can better zero-shot solve these puzzles, which serves as a non-trivial reference baseline. After training with $\mathcal{P}$uzzle$\mathcal{V}$erse (denoted as "w/ PV"), all LLMs shows significant improvement, particularly in reasoning and completeness. In compliance, all LLMs improve their scores by approximately 10% on average, with the gains being relatively modest due to the high baseline of compliance. The improvement in reasoning is particularly significant, with an average increase of over 150%. Completeness scores and machine metrics also see effective enhancement. In human evaluations, all LLMs show improved scores, with an average increase of over 100%. However, these LLMs still have a long way to go compared with GPT-4.

We also compare the performance of $\mathcal{P}$uzzle$\mathcal{V}$erse-trained LLMs with the agent mentioned in AgentBench (Liu et al., 2023) for the LTP task on our LTP dataset (denoted as "w/ AB"), as shown in Table 8. We adopt both creative metrics and machine metrics for evaluation. We find that $\mathcal{P}$uzzle$\mathcal{V}$erse achieves better results, with an average improvement of 40.5% over the agent. This improvement is likely because the agent can be considered an external prompt-based method, whereas our approach involves training, which better enhances LLMs' performance. There are also some frameworks related to lateral thinking that are not specifically designed for it. Therefore, we only qualitatively compared their target tasks, core technology, innovation, and performance in Table 7.

In addition, we evaluate $\mathcal{P}$uzzle$\mathcal{V}$erse-trained LLMs on other reasoning tasks, including story under-standing, story generation, and reading comprehension, as shown in Table 10. We use a one-shot evaluation method, providing each data point with one example. We find that $\mathcal{P}$uzzle$\mathcal{V}$erse-trained LLMs exhibit significant enhancements compared to vanilla models, highlighting the adaptability of $\mathcal{P}$uzzle$\mathcal{V}$erse across a range of reasoning tasks.

**Ablation Study.** After that, we adopt an ablation study to evaluate the contributions of each module within the $\mathcal{P}$uzzle$\mathcal{V}$erse framework. Due to the strong correlation between the creativity metric and the human metric, we primarily analyze these two metrics, as highlighted in Fig 4. Detailed results are shown in Tables 11. We observe it is evident that each module within the $\mathcal{P}$uzzle$\mathcal{V}$erse framework has a significant impact on lateral thinking. We can see that for all dimensions, the scores decrease when any single module is removed. Notably, removing the teacher-forcing module (denoted as "w/o TF") leads to the largest decline across various dimensions, indicating that the teacher-forcing module plays a crucial role in maintaining overall performance. The next most impactful module is reinforcement learning (denoted as "w/o RL"). Free-generation (denoted as "w/o FG") has the smallest effect across all dimensions, showing minimal decline when removed. For creativity and human evaluations, removing the teacher-forcing module results in substantial decreases in human

Table 11: Performance of training LLMs with $\mathcal{P}$uzzle$\mathcal{V}$erse variants which are removed a certain module. "w/o TF", "w/o RL", and "w/o FG stand for variants without teacher-forcing, RL, and free-generation, respectively.

| | Compliance | | | | | Creativity Reasoning | | | | | Completeness | | | | | Human - | | | | |
|---|---|---|---|---|---|---|---|---|---|---|---|---|---|---|---|---|---|---|---|---|
| | w/ PV | w/o TF | w/o FG | w/o RL | w/o PV | w/ PV | w/o TF | w/o FG | w/o RL | w/o PV | w/ PV | w/o TF | w/o FG | w/o RL | w/o PV | w/ PV | w/o TF | w/o FG | w/o RL | w/o PV |
| baichuan | 84.4 | **80.3** | 83.9 | 82.6 | 79.5 | 57.0 | **45.8** | 54.5 | 52.3 | 23.4 | 68.1 | **50.5** | 64.9 | 58.3 | 32.3 | 3.8 | **2.0** | 3.6 | 2.9 | 1.9 |
| MOSS | 84.1 | **78.2** | 83.5 | 82.0 | 76.0 | 56.0 | **47.2** | 55.7 | 53.5 | 20.5 | 67.4 | **52.7** | 66.5 | 62.4 | 31.4 | 3.6 | **2.1** | 3.3 | 2.5 | 1.7 |
| BELLE | 83.7 | **76.2** | 82.1 | 80.3 | 74.7 | 48.9 | **32.1** | 44.1 | 42.8 | 19.6 | 51.2 | **44.7** | 49.2 | 47.2 | 31.1 | 2.8 | **1.9** | 2.7 | 2.4 | 1.4 |
| ChatGLM | 83.6 | **73.5** | 81.6 | 77.8 | 72.6 | 46.0 | **31.5** | 42.6 | 39.2 | 17.8 | 51.1 | **40.6** | 49.9 | 46.8 | 29.5 | 2.9 | **1.5** | 2.4 | 2.1 | 1.2 |
| Average | 84.0 | **77.1** | 82.8 | 80.7 | 75.7 | 52.0 | **39.2** | 49.2 | 47.0 | 20.3 | 59.5 | **47.1** | 57.6 | 53.7 | 31.1 | 3.3 | **1.9** | 3.0 | 2.5 | 1.6 |
| ↓ | - | 6.9 | 1.2 | 3.3 | 8.3 | - | 12.8 | 2.8 | 5.0 | 31.7 | - | 12.3 | 1.8 | 5.8 | 28.4 | - | 1.4 | 0.3 | 0.8 | 1.7 |
| ↓(%) | - | 8.2 | 1.4 | 3.9 | 9.8 | - | 24.7 | 5.3 | 9.7 | 60.9 | - | 20.7 | 3.1 | 9.7 | 47.7 | - | 42.7 | 8.4 | 24.4 | 52.7 |

scores and reasoning, while compliance sees a smaller decline, likely due to its high baseline. These findings indicate that using the complete $\mathcal{P}$uzzle$\mathcal{V}$erse framework brings the greatest improvement across all metrics, highlighting its positive impact on enhancing LLMs' lateral thinking capabilities in problem-solving games.

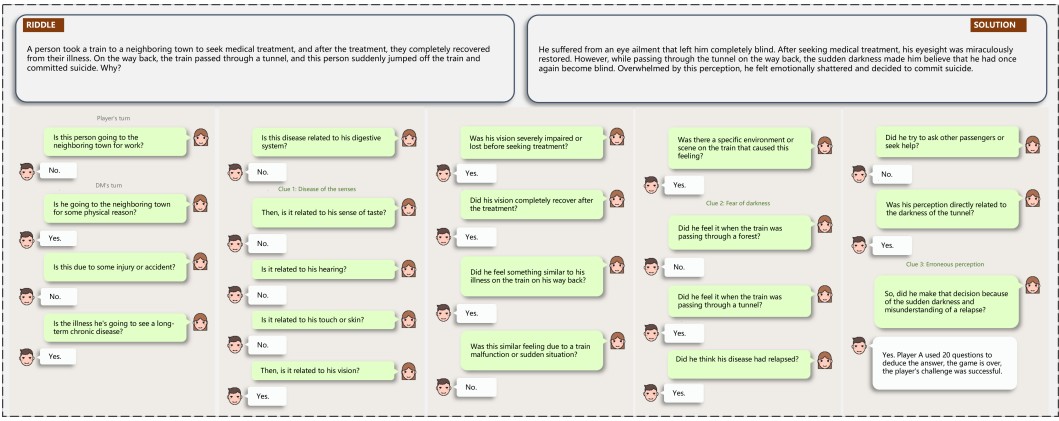

Figure 5: A good case of baichuan trained with $\mathcal{P}$uzzle$\mathcal{V}$erse on a lateral thinking puzzle.

**Case Study.** We analyze a good case as shown in Fig. 5. The LLM first asks about the reason of waking up, ruling out the possibility of thirst or hunger as the reason for waking up. Then it shifts the focus to health and asked, eliminating any association with studying or work. Next, the LLM continues to delve deeper, inquiring, "Do you have a certain disease that requires you to take medication at regular intervals?", further revealing that "sleep forever" has relationship with regular medication. Finally, the LLM asks whether not taking the medication on time threaten your life, confirming that not adhering to the medication schedule could endanger the life and therefore hinting at the cause of demise. Through these precise questions, the LLM successfully deduces that disease in the riddles requires regular medication, and failing to take it on time could be life-threatening. We showcase some bad cases in Table 12. The questions, such as whether the man checks the door lock or call the police, are indeed not directly relevant to the progression of the puzzle. After a series of answers with "Irrelevant", the LLM still asks some irrelevant questions.

Table 12: A bad case of baichuan trained with $\mathcal{P}$uzzle$\mathcal{V}$erse on a lateral thinking puzzle.

| Riddle | Solution | Questions and Answers | Clues |
|---|---|---|---|
| "Your takeout has arrived." "Okay." After the door closed, the man closed his eyes in terror. Shortly afterward, he experienced intense fear and anxiety. Let's reason this out. | The man lived alone and one night, feeling hungry, he ordered takeout but fell asleep while waiting. In the middle of the night, he was awakened by the sound of someone knocking on the door with the delivery. As he prepared to get out of bed, he heard someone's voice outside the door and realized that there was someone else in his home. Fearfully, he closed his eyes and pretended to be asleep. However, shortly afterward, he heard someone whispering in his ear, saying, "I know you're not asleep." | Question: Did the man check the door lock after hearing someone at the door? Answer: Irrelevant. Question: Does the man have surveillance cameras installed in his house? Answer: Not important. Question: Did the man lock the door after closing it? Answer: Irrelevant. Question: Did the man call the police after hearing someone at the door? Answer: Irrelevant. Question: Did the man close his eyes because of psychological fear? Answer: Irrelevant. | Takeout delivery Late at night Sounds at the door Closing the door Someone whispering in your ear |

# 5 Related Work

**Puzzle Solving.** For example, Jiang et al. (2023) introduced a multiple-choice QA task designed to test and benchmark the lateral thinking abilities of LLMs. Huang et al. (2023) proposed LatE

Zhao & Anderson (2023) focused on the ability of LLMs to solve and create puzzles in NPR Sunday Puzzles. King (2023) pointed out the challenges LLMs face in generating anagrams. Zhang et al. (2024) introduced a novel solver-layer adaptation (SoLA) method that enhances the puzzle-solving capabilities of LLMs. Wu et al. (2023) delved into the use of GPT-4 for tackling more complex mathematical problems. Xie et al. (2023) proposed OlaGPT to approximate various cognitive processes, including reasoning and decision-making. Sarathy et al. (2024) introduced ESCAPE using puzzle video games to study cognitive processes in creative problem-solving. Wang et al. (2024) player behavior in a puzzle game to identify effective problem-solving strategies. Differently, our research explore the potential of LLMs in lateral thinking within puzzle-solving games.

Although some work focus on lateral thinking puzzles and their application in evaluating LLMs, they only provides evaluations without offering solutions. For example, Jiang et al. (2023) introduced a multiple-choice QA task designed to test and benchmark the lateral thinking abilities of LLMs. Huang et al. (2023) proposed LatEval, an interactive benchmark that challenged LLMs on lateral thinking by assessing the quality of questions posed and the integration of information during problem-solving. Todd et al. (2024) explored the use of the "Connections" puzzle game as a benchmark for evaluating LLMs' abstract reasoning and semantic understanding. León Corrales et al. (2010) investigated how lateral thinking puzzles could enhance critical thinking and motivation in students' opinion paragraph writing, leading to improved writing skills. Lin et al. (2021) introduced a multiple-choice QA task focused on riddle-style questions that required commonsense reasoning and linguistic creativity, with a dataset of 5.7k examples. In contrast to these methods, we use LLMs for supervised fine-tuning and reinforcement learning, dynamically generating and optimizing question-posing paths, which significantly improved model performance on LTP tasks. Moreover, none of these benchmarks has as many samples as our work.

**Reasoning.** For example, Hao et al. (2023) utilized LLMs as world state predictors and strategic reasoners. Lu et al. (2023) introduced Chameleon in enhancing LLMs' compositional reasoning capability. Tarau (2023) automated deep reasoning in LLM dialog threads. Kıcıman et al. (2023) delved into causal reasoning capabilities of LLMs. Yoneda et al. (2023) introduced Statler to enhance LLMs' long-horizon reasoning capability in robotic tasks. Paranjape et al. (2023) presented ART to generate intermediate reasoning steps. Chen et al. (2023c) introduced ChatCoT by chain-of-thought reasoning. However, these work mainly focus on vertical thinking instead of lateral thinking.

**Question Generation.** For example, Chen et al. (2019) designed a reinforcement learning model for natural question generation. Tavares et al. (2023) delved into LLM strategies in generating questions on dialogue state tracking. Kai et al. (2021) proposed a double-hints method for visual question generation. Uehara et al. (2022) stressed the significance of sub-questions in enhancing primary visual queries. Arora et al. (2022) explored effective prompting strategies for LLMs. Abdelghani et al. (2022) harness GPT-3's capabilities in children's curiosity-driven questioning. However, these studies focus on reshaping question generation instead of searching valuable questioning points.

**Story Understanding.** For example, Yuan et al. (2022) introduced a platform fostering human-LLM story-writing collaborations. Swanson et al. (2021) unveiled STORY CENTAUR, optimizing LLMs for creative endeavors. Dong et al. (2022) spotlighted CoRRPUS to boost story consistency in LLM outputs. Bhandari & Brennan (2023) assessed the trustworthiness of LLM-generated children's stories. Chen et al. (2023b) advocated for LLMs to generate complex narratives. Lee et al. (2022) explored LLM-enabled interactive story rewriting. Méndez & Gervás (2023) utilized ChatGPT in narrative "sifting." Together, these contributions highlight the potential of LLMs in story generation and comprehension.

# 6 Conclusions and Future Work

In exploring the potential of LLMs, we've pinpointed their impressive aptitude for lateral thinking, which is instrumental for grasping intricate and nuanced contexts. By introducing the Lateral Thinking Puzzles and its complementary dataset, we illuminate the depth of this capability within LLMs. Our proposed $\mathcal{P}$uzzle$\mathcal{V}$erse framework is designed to further enhance LLMs' lateral thinking capabilities, and our proposed creativity metric offers a comprehensive evaluation. Experiments show the effectiveness of $\mathcal{P}$uzzle$\mathcal{V}$erse in not only LTP but also other reasoning tasks. Future research can delve into more intricate thinking scenarios and introduce the integration of multi-modal data, further enhancing LLMs' lateral thinking in puzzle-solving games.

# Ethic Statement

We analyze potential negative impacts and make ethic statement. Firstly, although lateral thinking encourages creativity and non-traditional solutions, these solutions may not align with societal norms or ethical standards in practical applications. Secondly, enhancing lateral thinking capabilities might exacerbate existing biases in LLMs. The previous training data for LLMs may already contain societal biases, and in lateral thinking tasks, these biases could be amplified or perpetuated through the generation of non-traditional solutions. To address these issues, we conduct a more comprehensive analysis of the societal impacts of these capabilities and explore how to incorporate stricter bias detection and correction mechanisms in model development and evaluation. Additionally, ethical reviews are integrated into the evaluation framework of model applications to ensure that the enhancement of lateral thinking capabilities does not lead to adverse societal consequences.

# Reproducibility Statement

Part of source code is available in https://anonymous.4open.science/r/haiguitang-EFA7/. We will open-source all data and code after being accepted. We make reproducibility statement on data construction as follows:

**Dataset Composition.** We constructed a novel lateral thinking puzzles dataset (LTP) to evaluate and enhance LLMs' lateral thinking capabilities in problem-solving games. Each puzzle includes a riddle with an unconventional solution, requiring creative, out-of-the-box thinking. We initially collected 647 Chinese lateral thinking puzzles from websites like Huiwan and used GPT-4 to generate additional puzzles with different semantics. These were carefully curated and expanded to maintain cultural nuances, resulting in a final dataset of 642,600 puzzles. Each puzzle includes questions, answers, and clues to guide LLMs towards the solution, evaluated for logical progression and safety. The comprehensive LTP dataset offers a robust framework for assessing and improving LLMs' lateral thinking abilities.

**Collection Process.** We constructed the Lateral Thinking Puzzles (LTP) dataset to enhance and evaluate LLMs' lateral thinking capabilities. Initially, we gathered 647 Chinese puzzles from websites like Huiwan. Using GPT-4, we generated additional puzzles with different semantics to ensure originality. Each puzzle includes a riddle and an unconventional solution, requiring creative thinking beyond traditional reasoning. To preserve cultural nuances, we focused on expanding the dataset in Chinese. We used GPT-4 to create sequences of yes-or-no questions, answers, and clues for each puzzle, designed to guide LLMs toward the solution. Both the puzzles and the question sequences were rigorously evaluated to ensure logical consistency and quality. To ensure safety, we filtered out puzzles with potentially harmful content. This meticulous process resulted in a high-quality dataset of 642,600 puzzles, providing a robust tool for assessing and improving the lateral thinking capabilities of LLMs in problem-solving games.

**Preprocessing/cleaning/labeling.** To ensure the quality and safety of the LTP dataset, we implemented a thorough preprocessing, cleaning, and labeling process. Initially, we used GPT-4 to generate additional puzzles, ensuring they mirrored the style of the collected Chinese puzzles but with different semantics. Each generated puzzle underwent rigorous evaluation to meet specific criteria, such as logical consistency and cultural relevance. Puzzles scoring below a threshold were discarded. Next, we created sequences of yes-or-no questions, answers, and clues for each puzzle, designed to guide the LLMs incrementally towards the solution. These sequences were evaluated for logical progression and accuracy, with inadequate sets being discarded. To maintain dataset integrity and minimize risks, we used GPT-4 to automatically detect and flag potentially unsafe content, such as detailed descriptions of violence or horror. Entries containing such content were removed. Manual rating by volunteers further ensured the dataset's quality, with puzzles scoring below a set threshold being excluded. The final dataset, comprising 642,600 puzzles, was thoroughly vetted for reliability and cultural nuance, ensuring it serves as a robust tool for enhancing LLMs' lateral thinking capabilities.

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

Table 13: Sample puzzles in LTP including riddles, solutions, questions, answers and clues.

| Riddle | Solution | Questions and Answers | Clues |
|---|---|---|---|
| One night, Xiao Ming made a phone call but hung up as soon as it was answered without waiting for the other person to speak. Why? | Xiao Ming was resting in a hotel when he couldn't fall asleep due to the loud snoring coming from the next room. So, he used the hotel's internal phone to wake up the person next door, and quickly fell asleep while they were no longer snoring. | Question: Is the phone the one Xiao Ming uses to make calls to others? Answer: Yes. Question: Is the phone the one Xiao Ming uses to call the hotel reception? Answer: Not important. Question: Is the phone the one Xiao Ming uses to call the neighboring room? Answer: Yes. Question: Is the phone the one Xiao Ming intentionally hangs up on the other person? Answer: Yes. Question: Is the phone the one Xiao Ming uses to create noise? Answer: Yes. Question: Is the phone the one Xiao Ming uses to complain about the noise to the neighboring room? Answer: Not important. Question: Is the phone the one Xiao Ming uses to request the neighboring room to quiet down? Answer: Not important. Question: Is the phone the one Xiao Ming uses to communicate with the neighboring room? Answer: Not important. Question: Is the phone the one Xiao Ming uses to disturb the sleep of the neighboring room? Answer: Yes. Question: Is the phone the one Xiao Ming uses to remind the neighboring room to stop snoring? Answer: Yes. | Resting in a hotel Snoring disrupts Waking up the neighbor Unable to fall asleep |
| The woman went to the library to borrow a book. When she opened it, she cried. | The woman is the author of this book. She inserted 100 yuan into the book and donated it to the library. After several years, she came back to borrow the same book, only to find the money still inside. This indicates that nobody has actually read her book. | Question: Is the book written by the woman herself? Answer: Yes. Question: Is the book borrowed by the woman from the library? Answer: Yes. Question: Is the book the one the woman borrowed from the library? Answer: Yes. Question: Is there a currency note in the book? Answer: Yes. Question: Why did the woman cry? Answer: Not important. Question: How much money did the woman put between the pages of the book? Answer: Not important. Question: Did the woman donate the book with the money inside to the library? Answer: Yes. Question: How long did it take for the woman to come back to borrow the book? Answer: Not important. Question: Is the money still inside the book? Answer: Yes. Question: Does the situation imply that nobody looked at the woman's book? Answer: Yes. | The woman borrowed a book she cried There was money inside the book She donated it to the library The money is still inside the book. |
| In a tall building at night, a woman was hanging clothes on the balcony. Suddenly, she unintentionally glanced at the building across from hers and was instantly horrified. | The woman saw an ongoing murder incident in the building across from hers, and the murderer also noticed her witnessing the event. The reason the woman was instantly horrified was that the murderer was counting the number of floors in her building. | Question: Is the woman hanging clothes out at night? Answer: Yes. Question: Is the woman in a high-rise building where she lives? Answer: Yes. Question: Is the woman hanging clothes on the balcony? Answer: Yes. Question: Did the woman accidentally look towards the building across the street? Answer: Yes. Question: Did the woman see something happening in the building across the street? Answer: Yes. Question: Did the woman witness an ongoing murder incident? Answer: Yes. Question: Did the killer notice that the woman witnessed his actions? Answer: Yes. Question: Did the woman feel terrified because she realized she had been discovered? Answer: Yes. Question: Is the killer counting the number of floors where the woman is located? Answer: Yes. Question: Does the number of floors where the woman is located have significance to the killer? Answer: Yes. | At night In a tall building The woman looked towards the building across instantly felt a chilling sensation The murderer was counting the number of floors. |
| A wealthy man made a phone call to his beloved wife, and as a result, she died. | In the wealthy man's house, a burglar entered. While the wealthy man was making a phone call, his wife was hiding in a certain place. Due to the phone not being on silent mode, the ringtone sounded and exposed the wife's location, leading to her being killed by the burglar. | Question: Was the wife at home when the millionaire called her? Answer: Not important. Question: What was the reason for the millionaire to call his wife? Answer: Not important. Question: Is the phone the one the millionaire used to call his wife? Answer: Yes. Question: Did a thief enter the millionaire's house? Answer: Yes. Question: Was the wife hiding somewhere when the millionaire made the phone call? Answer: Yes. Question: Did the wife's location get exposed after the phone rang? Answer: Yes. Question: Was the wife killed by the thief? Answer: Yes. Question: Was the wife killed because of the ringing of the phone? Answer: Yes. Question: Did the thief kill the wife because he knew her location? Answer: Yes. Question: Did the thief kill the wife after discovering her hiding place? Answer: Yes. | The wealthy man called his wife His wife died A burglar entered the house The phone's ringtone sounded The wife's location was exposed. |
| The painter received a phone call, and as he looked at a mermaid painting on the table, he suddenly started crying. | The painter is a single father, and because his son constantly asked about his mother, he told his son that the mother is the mermaid in the painting. The young son took it seriously and always said he wanted to go into the water to find his mother. Due to this situation, he was eventually sent to a mental hospital. The painter received a call from the mental hospital, informing him of his son's suicide by drowning. As he looked at the mermaid painting on the table, he deeply regretted not realizing his son's mental issues earlier or explaining the situation clearly, which ultimately led to his son's tragic suicide. | Question: Is the painter single? Answer: Not important. Question: Is the phone call the painter received an important event? Answer: Yes. Question: Did the painter create the mermaid painting he saw? Answer: Yes. Question: Does the painter's son believe that his mother is the mermaid in the painting? Answer: Yes. Question: Was the painter's son sent to a mental hospital because he was searching for his mother? Answer: Yes. Question: Is the phone call the painter received about his son? Answer: Yes. Question: Did the painter's son die from a suicide by drowning? Answer: Yes. Question: Does the painter regret not realizing his son's mental issues earlier? Answer: Yes. Question: Does the painter regret not explaining clearly about his son's mother? Answer: Yes. Question: Did the painter cry after seeing the mermaid painting on the table? Answer: Yes. | The painter received a phone call There was a mermaid painting His son went to a mental hospital His son died by suicide drowning The painter deeply regretted his past actions. |
| That painting depicted a man with sharp features, vividly lifelike. The next day, when I saw the painting again, I felt a tingling sensation on my scalp, and I couldn't utter a single word of praise. | I entered a rundown small hotel late at night. When I entered the room, even the light was broken, and the room was dimly lit. There was a painting on the opposite side of the bed, depicting a man with sharp features, vividly lifelike, just like the Mona Lisa. I always felt that the person in the painting was constantly watching me. It wasn't until the next morning, when it was bright outside, that I realized the supposed painting was actually a window. Last night, there was a man standing outside the window watching me, but due to the dim light, I mistook him and the window frame for a painting. | Question: Is the painting in a run-down small hotel? Answer: Yes. Question: Is the man in the painting very handsome? Answer: Yes. Question: Is the man in the painting depicted with clear features and lifelike appearance? Answer: Yes. Question: Does the man in the painting make the owner uncomfortable? Answer: Yes. Question: Was the painting later discovered to be a window by the owner? Answer: Yes. Question: Was the location of the window mistaken for a painting by the owner? Answer: Yes. Question: Did the owner feel that the lighting was dim when looking at the window at night? Answer: Yes. Question: Did the owner mistake the man standing outside the window for a painting? Answer: Yes. Question: Did the owner realize that the man on the window was continuously watching him? Answer: Yes. Question: Did the owner only discover that it was actually a window the next morning? Answer: Yes. | Rundown small hotel Man with sharp features Feeling uncomfortable Window The owner mistook it for a painting. |

