# OpenReview forum: "Do Large Language Models have Lateral Thinking in Puzzle-Solving Games?"
_ICLR.cc/2025/Conference — Submitted to ICLR 2025_

### Official Review · Reviewer_Pnux · 2024-10-30

**Soundness:** 3
**Presentation:** 3
**Contribution:** 2
**Rating:** 5
**Confidence:** 3

**Summary:**

The authors constructs the largest lateral thinking puzzle dataset by far as well as a novel set of metric to evaluate lateral thinking ability. They also proposes a PuzzleVerse framework that consists of SFT, RM, and RL. Extensive experiments are conducted to evaluate performance on the LTP dataset as well as to evaluate performance in other similar tasks like story generation.

**Strengths:**

1. The authors nicely justify the importance of lateral thinking as a crucial ability for LLM reasoning. The paper is well-written and clarifying.
2. The author novelly yet carefully curated a large set of lateral thinking puzzles which can effectively measure lateral thinking abilities. They also proposes a comprehensive set of creativity metrics for evaluation.

**Weaknesses:**

1. The dataset is only available in Chinese due to a loss of cultural context during translation. This limits the use case of using this dataset for more extensive comparison of LLM reasoning capability as cultural context will be crucial for solving puzzles in this dataset (for example models trained using English dataset would not understand "square dancing"). I would suggest the authors to develop a culture-neutral subset.
2. The evaluation dataset chosen outside of the LTP dataset seems debatable. I would not really consider story understanding or reading comprehension task to be using lateral thinking. One immediate way to improve this is simply evaluating the framework on previous LTP dataset.

**Questions:**

1. For baseline evaluations the authors choose a zero-shot setting. I am curious why experiments with few-shot settings are not done? The dataset is novel as puzzles like these are not commonly seen but I would assume that these puzzles follows some intrinsic patterns as of how the solution is derived from the question. In other words in zero-shot settings the model might not grasp what kind of questions are good to ask but this problem is instantly solved in few-shot settings (similar to how human would quickly get better in "HaiGuiTang").
2. (This question might be somewhat vague and I'm not being critical just curious to see what the authors think) How does the author justify the idea of language models even being ABLE to do lateral thinking? The training objective of LMs naturally leads to models selecting the most possible outcomes so I would be surprised to see LLMs thinking out of the box to such extreme extent as shown in these puzzles.

---

### Official Review · Reviewer_odtQ · 2024-11-01

**Soundness:** 2
**Presentation:** 2
**Contribution:** 2
**Rating:** 3
**Confidence:** 4

**Summary:**

To test and enhance lateral thinking in LLMs, the paper introduces a large dataset called Lateral Thinking Puzzles (LTP), composed of riddles with unconventional solutions. It also proposes a framework, PuzzleVerse, which guides LLMs in incrementally questioning and deducing answers through yes-or-no questions, designed to stimulate creative problem-solving strategies. In experiments, LLMs trained with PuzzleVerse demonstrate significant improvements in solving puzzles creatively, thus providing a new perspective to reasoning.

**Strengths:**

1. Lateral thinking promotes creative reasoning in LLMs, helping them move beyond straightforward logical solutions and explore unconventional answers, which could be valuable for complex problem-solving.
2. The performance of the framework is good.

**Weaknesses:**

1. The paper asserts that its approach significantly enhances the creativity of LLMs by extending the scope from text-based riddles to a broader category of puzzles. However, this claim might be overstated.
2. The dataset and framework's aim is commendable in seeking to bolster LLM creativity through lateral thinking. However, the use of clues in the SFT and RL training processes seems to contradict this goal. By providing clues, there's an implicit guidance that may limit the LLMs' ability to explore solutions outside of the predefined parameters.

**Questions:**

1. The paper mentions that the dataset is designed with a focus on the Chinese language. However, the inclusion of GPT-4 in the benchmark raises a question regarding its suitability. Given that GPT-4 is known for its superior performance in English, it would be beneficial for the paper to discuss the rationale behind incorporating a model that excels in a different language context. This discussion could provide insights into how the model's strengths in English might influence the results within a Chinese-centric dataset or whether there are specific reasons for expecting GPT-4 to perform well despite the language discrepancy.
2. When assessing the performance of various LLMs on the dataset, it is crucial to consider the impact of model size and complexity. The paper compares the performance of different LLMs but does not explicitly mention the number of parameters for each model. Model performance can be significantly influenced by the number of parameters, which affects their capacity for learning and generalization. It would greatly enhance the analysis if the paper could provide details on the parameter count for each model included in the comparison.

---

### Official Review · Reviewer_gKov · 2024-11-03

**Soundness:** 3
**Presentation:** 3
**Contribution:** 3
**Rating:** 6
**Confidence:** 4

**Summary:**

The paper explores the lateral thinking abilities of LLMs in puzzle-solving scenarios, where solutions require creative, non-linear approaches. The authors introduce the “Lateral Thinking Puzzles” dataset. It includes unconventional riddles designed to test LLMs' lateral thinking. They propose a framework, PuzzleVerse, to improve LLMs' performance on these tasks through question generation and reinforcement learning.

**Strengths:**

1. Novel Lateral Thinking Puzzles Dataset: The paper introduces the largest lateral thinking puzzle dataset. Each puzzle includes a riddle, unconventional solutions, a sequence of yes-or-no questions, answers, and clues. The dataset is carefully constructed to capture the nuances of lateral thinking and is validated through both automated and manual review processes to ensure high quality and coherence.
2. The PuzzleVerse framework combines supervised fine-tuning with reinforcement learning, utilizing a reward model that ranks questions based on relevance and coherence with the puzzle solution.
3. Experiments demonstrate significant performance gains, with LLMs achieving an average improvement of 101.9% after PuzzleVerse training on lateral thinking tasks. These results are benchmarked against powerful LLMs like GPT-4, providing a robust comparison.

**Weaknesses:**

1. LLM-judge/metrics might be a good help other than only relying on human evaluation. BLEU/ROUGE is not useful here.
2. The data creation part is not quite convincing since the challenging puzzle is not a easy task to generate. Some evaluation or human quality check might be needed.
3. Language is Chinese only.

**Questions:**

1. How to make sure the qulity of GPT-4 generated puzzles? Since these puzzles are quite challenging to GPT-4. With in-content learning, GPT-4 is able to creatively create new puzzles? Any evaluation on the qulaity?

---

### Official Review · Reviewer_44N2 · 2024-11-04

**Soundness:** 1
**Presentation:** 3
**Contribution:** 2
**Rating:** 3
**Confidence:** 3

**Summary:**

This paper contributes a new dataset of Lateral Thinking Puzzles for training and evaluation of the lateral thinking abilities of LLMs in the Chinese language. They further introduce the Puzzleverse framework where LLMs are instruction fine-tuned and aligned with a reward model on 70% of the dataset. Training with Puzzleverse shows improved performance in the created dataset and other reasoning tasks.

**Strengths:**

S1: The authors propose an automated synthetic data generation approach for evaluating and inculcating lateral thinking in LLMs
S2: The generated dataset is significantly larger than previous works

**Weaknesses:**

W1: The GPT-4 model is used to create, and evaluate the quality, consistency and correctness of most of the data limiting the upper bound of the performance of any model trained on this data to the GPT-4 model. Previous work [1] shows that even the GPT-4 model performs poorly on lateral thinking limiting the potential of this dataset.

W2: There is no human verification of whether the puzzles included in the dataset created using GPT-4 can actually be solved. There is no human performance on the test set reported.

W4: During inference, there's a 70:30 split of the training set. Since a large amount of data is generated using an LLM there could be significant overlap between questions across the dataset.

W5: In a setting like Lateral thinking, an LLM's performance might differ a lot if evaluated multiple times on the same question. There are no variance studies or standard errors across multiple trials reported.

W6: Only 30% of the total data validated for correctness by humans. Within this filtered data - "Puzzles scoring below 6 are discarded, resulting in a final average score of 6.65" The justification for this threshold is unclear, as the questions should absolutely satisfy all those conditions for the puzzle to be lateral thinking puzzle. Furthermore the exact distribution of the scores is missing.

W6: The models with and without puzzleverse are not evaluated on existing lateral thinking datasets like [1].


[1] LatEval: An Interactive LLMs Evaluation Benchmark with Incomplete Information from Lateral Thinking Puzzles

**Questions:**

Q1: What was the human score distribution for the 30% of data that was validated on those 8 metrics?

Q2: Could you elaborate the choice of threshold in validating the data?

Q3: What percentage of those 30% data was invalidated due to significant harmful content by humans? A similar fraction of such harmful content could still be a part of the remaining 70% remaining data.

Q4: Do the mentioned LLMs perform perfectly on those 647 original chinese puzzles? If not they could be used to test the generalizability of the puzzleverse framework.

Q5: Is the 30% data from test split the samples that were validated by volunteers?

**Details Of Ethics Concerns:**

- Possibly harmful content in the part of the dataset not validated by humans
- 3 Volunteers reviewed around 194,100 examples (30% of the total 642,700.) That's a significant time investment on the part of volunteers without compensation

---

### Meta-Review · Area_Chair_4vkk · 2024-12-21

**Metareview:**

This paper propose a new dataset of lateral thinking and designs a new framework that does reward fine-tuning with this dataset to promote LLM's creative thinking. Main concerns are two-folded. First, the dataset creation procedural is not quite convincing. Does the dataset truly reflect LM's lateral thinking ability remains debatable. Second, the framework to improve LLM's ability on this dataset is derivative. And whether it is true improvement or some degree of overfitting is not warranted.

**Additional Comments On Reviewer Discussion:**

No rebuttal is provided.

---

### Decision · Program_Chairs · 2025-01-22

Reject